# Beyond Vegetation: A Review Unveiling Additional Insights into Agriculture and Forestry through the Application of Vegetation Indices

Sergio Vélez [1,*](ID), Raquel Martínez-Peña [2] and David Castrillo [3,*](ID)

1    Information Technology Group, Wageningen University & Research, 6708 PB Wageningen, The Netherlands
2    Regional Institute of Agri-Food and Forestry Research and Development of Castilla-La Mancha (IRIAF), CIAG-"EL CHAPARRILLO", Ctra. Porzuna km 4, 13071 Ciudad Real, Spain
3    Regulatory Council for Organic Farming of the Principality of Asturias (COPAE), Avda. Prudencio Glez. 81, Posada de Llanera, 33424 Asturias, Spain
*    Correspondence: sergio.velezmartin@wur.nl (S.V.); david@copaeastur.org (D.C.)

**Abstract:** Vegetation indices (VIs) have long been a crucial tool for monitoring plant growth and health, assessing the impact of environmental factors on vegetation, and supporting decision-making processes in agriculture and forestry. Traditionally, these mathematical formulations, leveraging the spectral response of plants to sunlight, have been instrumental in assessing vegetation health. However, emerging research suggests some unconventional applications that extend the scope of VIs. This article surveys the traditional and novel uses of VIs in various fields, including other applications such as climate change studies, disaster management, or microorganism assessment. It underscores the value of VIs, such as the normalized difference vegetation index (NDVI), or the enhanced vegetation index (EVI), in tracking climate-related shifts, mitigating the impact of disasters, assessing microbial terroir, organic production, and even aiding the wine industry. Nonetheless, despite their promise, the practical application of VIs also presents interpretational and technical challenges that need to be addressed. Hence, while the vast potential of VIs is underscored in this article, it also calls for continued technological advancements and research efforts to fully harness their benefits, ultimately aiding in the sustainable management of our environment. The objective of this review is not only to reflect on the current situation, but also to explore innovative and unconventional applications of vegetation indices. This involves anticipating the potential of this dynamic and rapidly advancing scientific tool, rather than solely following mainstream approaches.

**Keywords:** precision agriculture; satellite; microbial terroir; NDVI; canopy structure; vegetation indices; soil reflectance; crop health monitoring; UAV; multispectral; organic production

## 1. Introduction

In the fields of agriculture and forestry, the accurate and detailed assessment of vegetation health, structure, and composition has traditionally been an intricate process. For decades, researchers and practitioners have sought to develop effective methods to identify, categorize, and monitor the vast array of vegetative species and states that exist within these disciplines. The necessity of a robust mechanism to map and evaluate the diverse flora in these sectors was paramount, given their significant role in global food production and maintaining ecological balance. This requirement propelled the development and utilization of advanced scientific techniques. One of the most powerful tools to emerge from these efforts is vegetation indices (VIs), mathematical combinations of spectral bands designed to enhance the contribution of specific information captured by remote-sensing instruments [1]. These VIs have changed the way that vegetative data are perceived and interpreted, allowing for precise and consistent measurements on a large scale. By integrating information from different parts of the electromagnetic spectrum, VIs

can accentuate certain characteristics of vegetation that would be difficult or impossible to discern with the naked eye.

VIs have been widely used in agriculture and forestry for decades. They are remote-sensing tools that provide information about vegetation health and productivity by measuring the reflectance of light from plants. VIs are calculated using the spectral bands of satellite or airborne sensors, and they are used to estimate biophysical parameters such as leaf area index (LAI), biomass, chlorophyll content, and water content [2]. These VIs have proven instrumental in offering insights into the complex world of vegetation, transcending our conventional understanding, and ushering in an era of comprehensive, quantifiable vegetation analysis [3].

The importance of VIs in agriculture and forestry cannot be overstated, they play a pivotal role in precision agriculture, facilitating data-driven decisions that can enhance crop management, improve yields, and increase profitability by providing a straightforward and reliable assessment of the condition and health of crops, allowing for the monitoring of various aspects of plant growth and development such as chlorophyll content, leaf area, canopy structure, and water status [4]. This real-time monitoring capability, combined with the ability to assess wide tracts of land remotely, allows for swift intervention, improving both resource efficiency and sustainability. Therefore, the utilization of VIs has become widely recognized for its ability to measure, monitor, and analyze vegetation in various contexts, from assessing crop health to estimating forest-canopy density. Past research has extensively highlighted their efficiency and reliability. For instance, the normalized difference vegetation index (NDVI), arguably the most recognized VI, has been successfully used in large-scale drought monitoring, assessing forest-fire severity, and identifying disease stress in crops. Yet, this is just a glimpse into the myriad applications that VIs possess and due to their success, hundreds of VIs exist today [5]. Given the variety of VIs available, selecting the most appropriate indices to use can be challenging. Research has shown that multiple VIs may be necessary to best capture agricultural crop characteristics at different growth stages and under varying management practices [6]. Additionally, the continued development of remote-sensing methods, including the introduction of narrow-band or hyperspectral sensors, is opening up new possibilities for the use of VIs. Advancements in artificial intelligence and machine learning also present opportunities to enhance the predictive capabilities of these indices, adding another layer of sophistication to vegetation monitoring and analysis.

The field of remote sensing has seen a significant upsurge in research interest, with over 309,377 entries found in the Scopus database (Figure 1). This interest has primarily spilled over to the application of VIs, a common utilization of remote sensing, accounting for over 60,930 entries in the same database. Remote sensing encompasses a wide array of tools, including satellites and unmanned aerial vehicles (UAVs), each with distinct advantages. Traditionally, satellites have commanded substantial interest in the domain of VIs, a trend originating in the 1970s. However, since 2010, and particularly after 2015, there has been a surge in the usage of UAVs for employing VIs. This burgeoning use of UAVs has significantly closed the gap between these platforms and traditional satellites, indicating a potential paradigm shift in the field of remote sensing. The growing adoption of UAVs can be attributed to their lower operating costs, greater accessibility, and the flexibility to conduct more localized and precise surveys.

Nevertheless, these instruments are not just capable of reading vegetation characteristics; they also carry the potential to extract an extensive range of environmental, agricultural, and forestry parameters that remain relatively unexplored. Despite the wealth of research on the use of VIs in vegetation monitoring, there is a burgeoning field of study that seeks to delve beyond traditional applications, unveiling additional and novel insights into agriculture and forestry through the use of VIs. This review aims to shed light on these less-explored applications, emphasizing the relevance and potential of VIs in providing valuable information in contexts such as climate-change studies, disaster management,

microorganism detection, and yeast biodiversity—areas closely linked to agricultural and forestry practices but not solely reliant on vegetation measurements.

As researchers delve deeper into the vast realm of VIs and their applications, it becomes increasingly evident that they are only at the brink of understanding their full potential. Even though the use of VIs for vegetation analysis is well established, exploring their application in non-traditional domains presents an exciting frontier. Ultimately, this review intends to inspire a broader perspective on VIs, encouraging researchers, practitioners, and policymakers alike to consider these versatile tools' myriad potentials beyond conventional vegetation analysis. The objective is not just to reflect on where researchers stand today but also to explore innovative and unconventional applications of VIs. This involves anticipating the potential of this dynamic and rapidly advancing scientific tool, rather than solely following mainstream approaches.

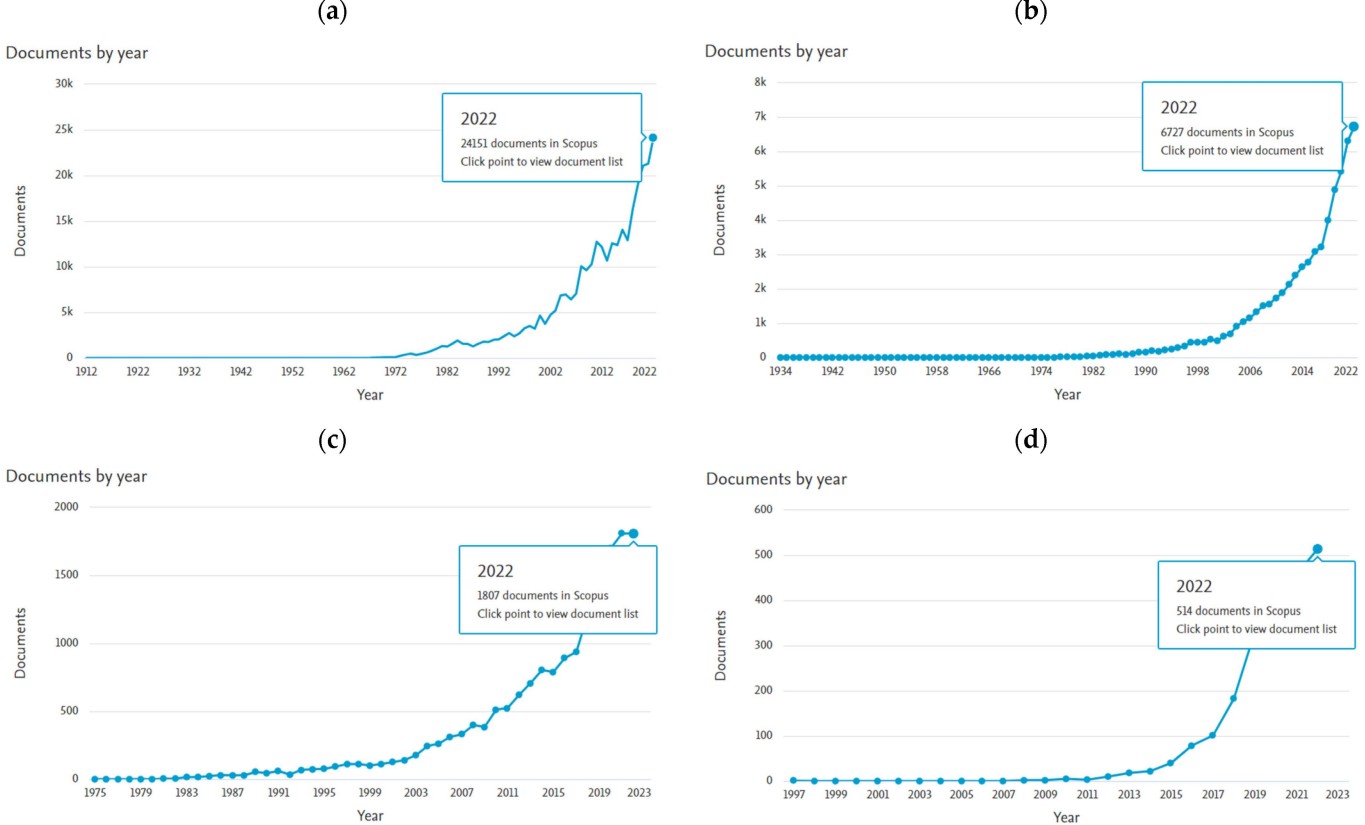

**Figure 1.** Results in Scopus database for (**a**) "Remote" AND "Sensing", (**b**) "Vegetation" AND "Index", (**c**) "satellite" AND "vegetation" AND "index", and (**d**) "drone" OR "UAV" AND "vegetation" AND "index".

## 2. Methodology

The study is a review of literature sourced from the Scopus database, one of the most extensive bibliographic databases for research literature. The research methodology was designed to ensure a systematic and in-depth analysis of the selected studies. The process was initiated by focusing on four key research words associated with remote sensing and vegetation indices (VIs). For this, four distinct search queries were deployed: (i) "Remote" AND "Sensing", (ii) "Vegetation" AND "Index", (iii) "satellite" AND "vegetation" AND "index", and (iv) "drone" OR "uav" AND "vegetation" AND "index". The results from this first step of the search are presented in Figure 1.

Next, attention was turned to the normalized difference vegetation index (NDVI) and the Scopus database was searched exclusively for entries related to "NDVI". The results of this search are presented separately in Figure 2.

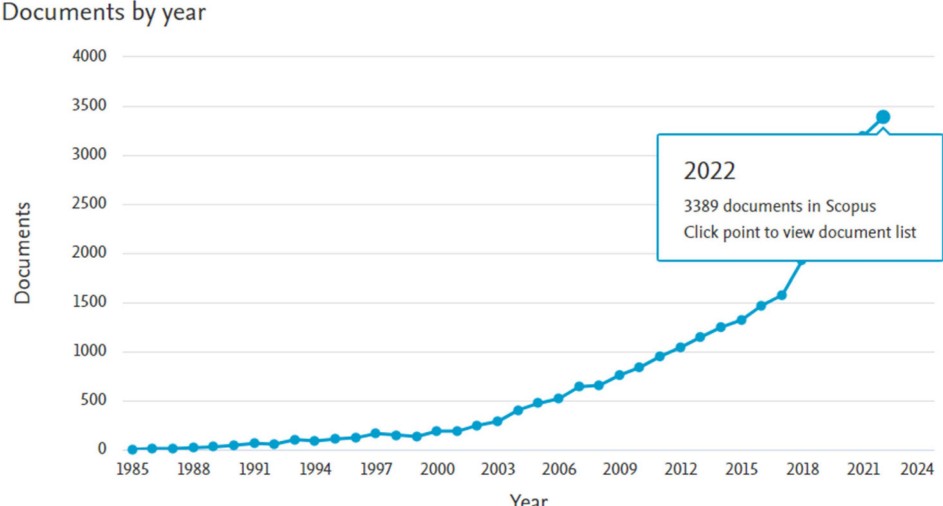

**Figure 2.** Results in Scopus database for "NDVI".

Further refining the investigation, a delve was made into other common VIs, leading to a search on (i) "EVI", (ii) "SAVI", (iii) "NDRE", and (iv) "CWSI". The findings from these searches are illustrated in Figure 3.

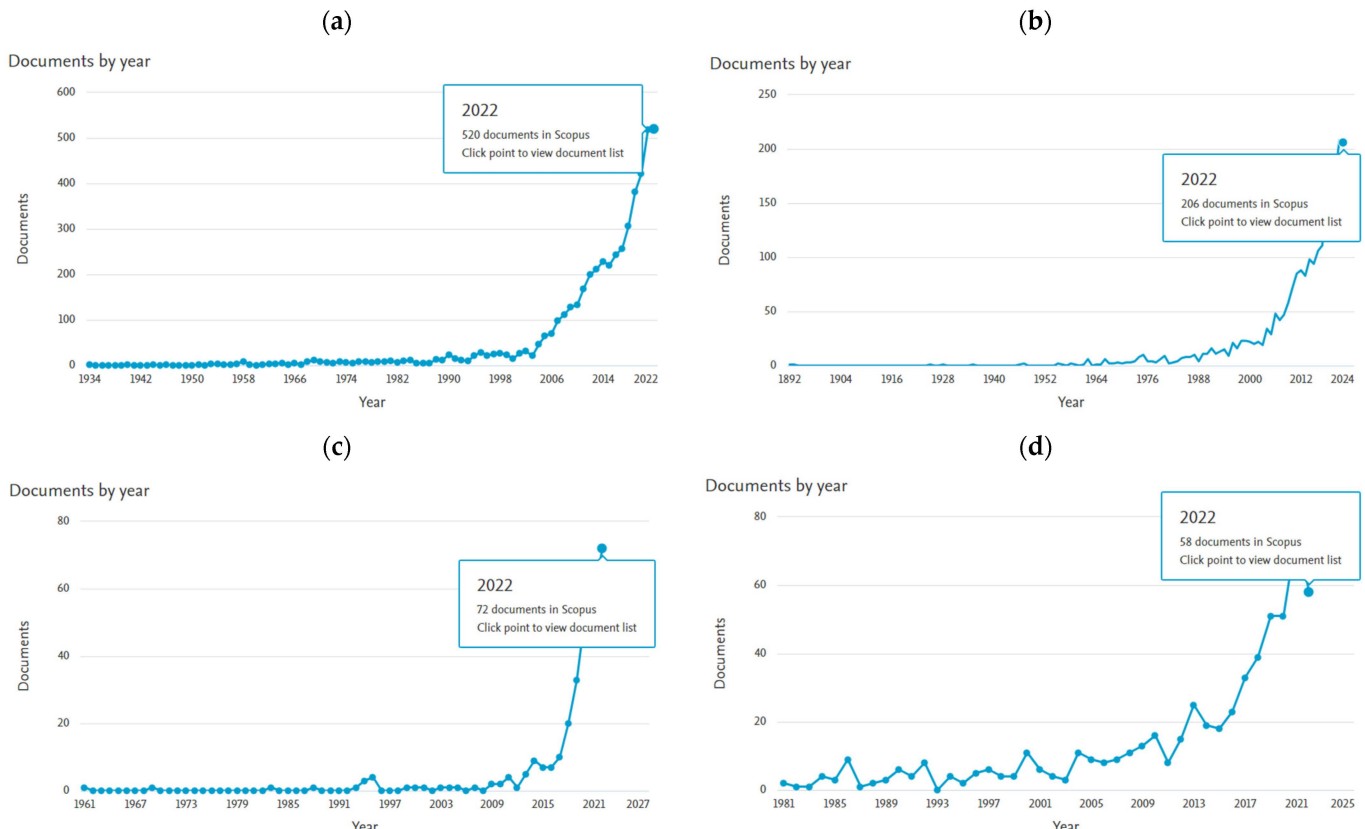

**Figure 3.** Results in Scopus database for (**a**) "EVI", (**b**) "SAVI", (**c**) "NDRE", and (**d**) "CWSI".

Lastly, to understand the wider implications of theNDVI beyond vegetation, specific searches were carried out linking the NDVI with other relevant domains. These search strings included: (i) "NDVI" AND "vegetation", (ii) "NDVI" AND "climate" AND "change", (iii) "NDVI" AND "disaster", and (iv) "NDVI" AND "microbial". The results of these targeted searches are documented in Figure 4.

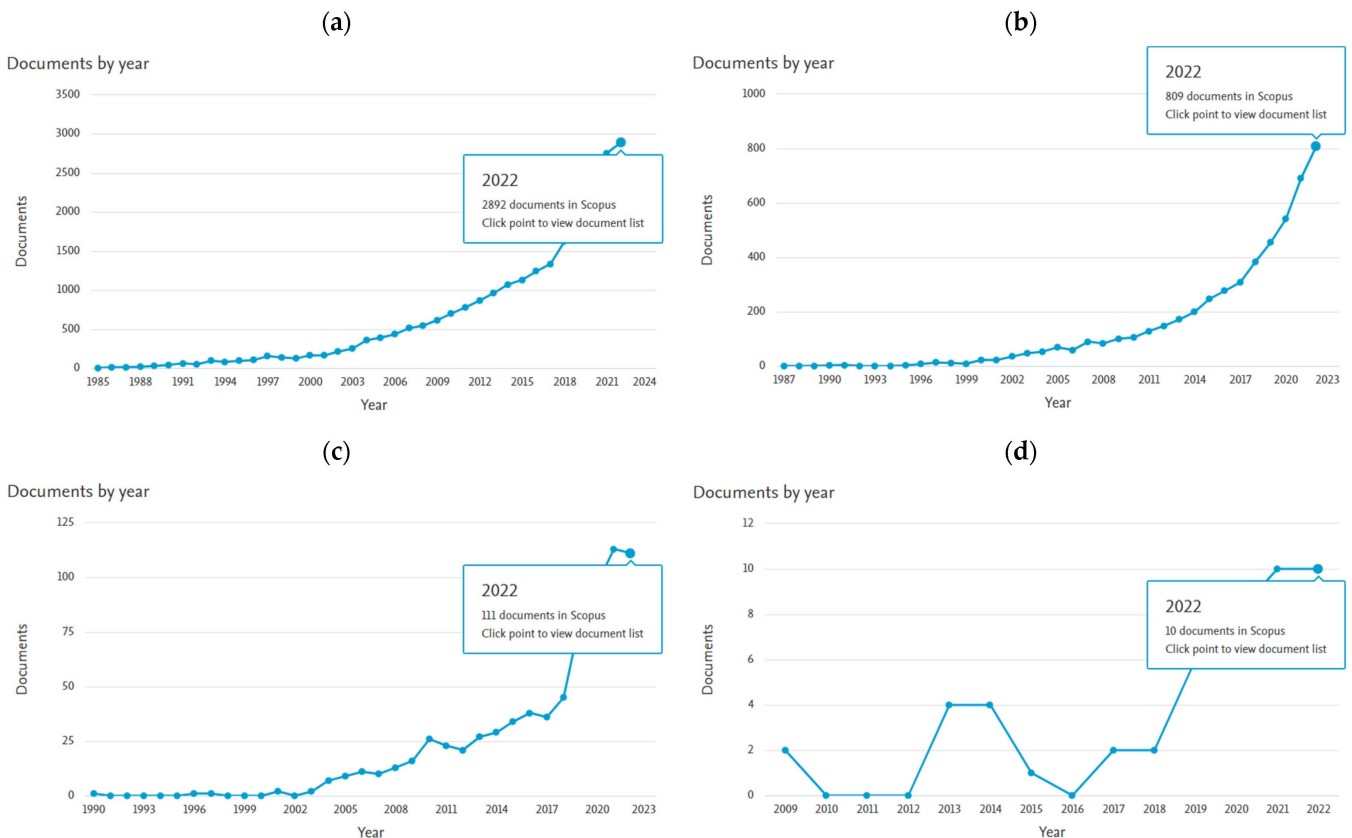

**Figure 4.** Results in Scopus database for (**a**) "NDVI" AND "vegetation", (**b**) "NDVI" AND "climate" AND "change", (**c**) "NDVI" AND "disaster", and (**d**) "NDVI" AND "microbial".

In order to maintain a reasonable volume of information, for each category the first 100 papers were analyzed, first by title and, if relevant for the interests of this review, the abstract and finally the full text, including references to try to find unconventional applications for VI. This structured methodology allowed the generation of an overview of the current state of knowledge related to the use of VIs in various fields, well beyond direct vegetation assessment.

## 3. Understanding Vegetation Indices

Vegetation indices are mathematical formulae that use the ratio of different wavelengths of light reflected by plants to estimate various vegetation characteristics. They are used in agriculture and forestry to monitor plant growth and health, as well as to assess the impact of environmental factors on vegetation. Several types of VIs exist, each with its own advantages and limitations. The concept of a vegetation index hinges on the distinctive ways that vegetation interacts with light. When sunlight strikes a plant, certain wavelengths are absorbed for photosynthesis, particularly in the blue (around 450 nm) and red (around 660 nm) parts of the spectrum. Conversely, the green (around 550 nm) and near-infrared (NIR, around 800 nm) light is reflected [7]. This characteristic spectral response is captured in the form of VIs.

VIs have become an indispensable tool in the fields of agriculture and forestry, providing a quantitative means of evaluating and monitoring plant health and growth [8]. VIs have traditionally been used to monitor crop and forest health and phenological changes, and to estimate biomass, providing vital information for managing agricultural and forestry resources [9]. In the realm of agriculture and forestry, they offer valuable information about plant health, growth, and productivity. These indices, such as the normalized difference vegetation index (NDVI), the soil-adjusted vegetation index (SAVI), and the enhanced vegetation index (EVI) among others, have been used to monitor and assess various aspects

of vegetation health [10]. The key principle behind VIs lies in the distinct spectral response of plant vegetation in comparison to other natural materials, such as soil and water [11]. In this way, research has been conducted on a variety of VIs, each serving a unique purpose and offering different advantages depending on sensor type and field conditions [4].

The normalized difference vegetation index (NDVI), one of the most commonly used VIs, leverages this differential absorption and reflection. It is computed using the formula: NDVI = (NIR − Red)/(NIR + Red) [12]. NDVI values range between −1 and +1, with higher values indicating greater vegetation density and health [13]. The NDVI is particularly beneficial for monitoring large agricultural areas and forest canopies due to its sensitivity to chlorophyll content, but can be limited by soil and atmospheric noise, and saturation in high biomass areas [14]. Moreover, the NDVI has shown its potential for establishing sampling methodologies within the field [15], for crop management and zoning in accordance with crop vigor [16], for mapping phenology metrics [17], and for estimating wheat grain yields [18]. This has led to a growing interest in this specific VI (Figure 2), with more than 28,000 entries until 2022 in the Scopus database, surpassing the numbers of other VIs such as the EVI, SAVI, NDRE, and CWSI, whose combined entries do not reach this number (Figure 3).

To circumvent some of the limitations of the NDVI, other indices such as the soil-adjusted vegetation index (SAVI) have been developed. The SAVI introduces a soil brightness correction factor (L) into the NDVI formula to minimize soil noise, making it useful in areas with sparse vegetation [19]. The formula for the SAVI is: SAVI = [(NIR − Red)/(NIR + Red + L)] × (1 + L). The normalized difference red edge (NDRE) index is another important VI that has been specifically designed to be sensitive to chlorophyll content in higher biomass areas, where the NDVI tends to saturate. The NDRE is calculated using the formula: NDRE = (NIR − RedEdge)/(NIR + RedEdge), where Red-Edge refers to the spectral band around 730 nm [20]. The enhanced vegetation index (EVI) is another modification of the NDVI that aims to optimize the vegetation signal in high biomass regions and improve sensitivity to atmospheric and canopy background conditions. The formula for the EVI is: EVI = G × [(NIR − Red)/(NIR + C1 × Red − C2 × Blue + L)], where G, C1, C2, and L are constants [14].

The canopy chlorophyll content index (CCCI) and the crop water stress index (CWSI) are indices that provide more specific insights. The CCCI, calculated as NDRE/NDVI, is a measure of the chlorophyll content per unit area of the canopy [21,22] and uses reflectance in the near-infrared (NIR) and red spectral regions to compensate for changes in canopy density. Its significance lies in its ability to detect relative changes in canopy chlorophyll or nitrogen content, making it a useful tool for monitoring plant health and fertility [23]. Meanwhile, the CWSI, calculated based on canopy temperature and atmospheric conditions, estimates water stress in crops and is a measure of the relative transpiration rate occurring from a plant at the time of measurement. It is calculated by using a measure of the canopy temperature of a plant (TC) and the vapor pressure deficit (VPD), which is a measurement of the dryness of the air [24,25]. Given its basis in canopy temperature and transpiration rates through baselines [26], the CWSI is a powerful tool for assessing water stress in crops, and thus for informing irrigation planning and management.

Finally, the visible atmospherically resistant index (VARI) is another noteworthy index designed to emphasize vegetation in the visible portion of the spectrum while mitigating illumination differences and atmospheric effects. It is particularly useful for RGB or color images and utilizes all three color bands [27]. The equation to calculate VARI is as VARI = (Green − Red)/(Green + Red − Blue) where green, red, and blue refer to the pixel values from the corresponding color bands of an image.

The different VIs, as described above, have their specific advantages and limitations. They offer a range of tools for measuring different aspects of vegetation, each with unique applications in agriculture and forestry. However, it is crucial to note that the performance and usefulness of these indices can be influenced by a multitude of factors, such as the type of vegetation, geographical location, time of year, and atmospheric conditions. Additionally,

the application of these indices requires careful calibration and interpretation, given the complexity of the processes they are designed to represent. Furthermore, despite the evident utility of these indices for vegetation assessment purposes, a review of the literature reveals a gap in research with respect to exploring non-traditional uses of VIs in forestry and agriculture. For instance, studies investigating the role of these indices in climate-change studies, disaster management, and the study of microorganisms and yeasts in relation to agriculture are lacking.

## 4. Unlocking New Horizons: Leveraging Vegetation Indices for Innovative and Diverse Applications beyond Vegetation Monitoring

In recent times, remote-sensing technology has experienced an upsurge in its applications, extending beyond the conventional scope of vegetation monitoring (Table 1). New studies have expanded the use of these indices, applying them in novel contexts beyond vegetation assessment within agriculture and forestry, such as the assessment of habitat conditions for wildlife conservation [28] or the effective detection of aquatic plants [29]. This showcases the versatility of VIs in research domains other than the conventional. Some of the novel domains encompassing the use of remote sensing include studies on climate change, disaster management, and microorganism assessment, among other applications that are not directly associated with vegetation assessment. On the other hand, certain applications such as quality assessment are more intrinsically linked to plant health and vegetation status and have been more extensively utilized.

**Table 1.** Vegetation indices for innovative and diverse applications beyond vegetation monitoring.

| Innovative Applications | Articles and Use of Vegetation Indices |
|---|---|
| Climate change | Significant indicators of climate-change effects on terrestrial ecosystems [9] <br> Relationship with climate factors such as precipitation [30] <br> Correlation with temperature and precipitation trends and the NDVI as an indicator of climate change [31] <br> Track climate-change impacts on crop phenology and productivity [32] <br> Proxy to assess changes in plant phenology and productivity in response to climate change [13] <br> Assess the effects of climate change in the Amazon Basin [33] <br> Mapping soil moisture in cultivated agricultural areas [34] <br> Assess atmospheric particulate pollution [35]. |
| Organic production | Agricultural management geared towards enhancing the yield quality in organic production [36,37] <br> Detect phytochemicals in organic agriculture to facilitate the adherence to certification audits, ensuring the maintenance of safe pesticide thresholds in conventional agricultural practices [38–40] <br> Increase the traceability of organic production [41] |
| Disaster management | Detect flood-affected areas [42] <br> Identify areas affected by wildfires [43] <br> Assess and monitor regional droughts [44] <br> Rapid damage assessment post-disaster [45] <br> Aid in monitoring and mapping wildfire damages and post-fire recovery [46] |
| Microorganisms and yeasts | Detect fungal infections in crops [47,48] <br> Monitor the induction of plant defense mechanisms [49] <br> Study the interaction of climate, topography, and soil properties with cropland [50] <br> Identify the presence of infections in plants [51] <br> Monitor microbial terroir and yeast-species richness within the vineyards [52] <br> Differentiate yeasts according to their fermentative capacity and a decision-making resource for designation of origin (DO) regulators and viticulturists [53] <br> Detect the presence of aggressive soil pathogens [54] |

**Table 1.** *Cont.*

| Innovative Applications | Articles and Use of Vegetation Indices |
| --- | --- |
| Quality assessment | Enhance wine-production management and productivity by providing insights into grape-quality variables [55]<br>Zoning according to vigor and quality parameters in grapes and wine [16]<br>Relationships with crop quality in cereals [56] |
| Leaf area and photosynthetically active radiation (FPAR) calculation | Strong linear relationship between the satellite-derived NDVI time series and the leaf area of the crop [57]<br>Estimating corn LAI using hyperspectral reflectance data [58]<br>Determine the radiation intercepted by the plant to estimate the LAI [59]<br>Regional-scale method for accurately estimating rice LAI during the growing period [60]<br>LAI estimation in semi-arid grasslands [61]<br>Study of the trade-off between the scale of the research and the availability of data [62]<br>Vegetation indices other than the NDVI to improve LAI estimations [63] |

However, when examining the scope of research, entries related to the NDVI and vegetation still vastly outnumber those associated with these recent concepts. Despite this disparity, a clear upward trend is observed in some areas, such as the articles related to the NDVI and climate change. Nevertheless, some entries, like those related to microorganisms, are considerably fewer in number (Figure 4).

### 4.1. Climate Change

Climate change represents one of the most profound challenges facing the agricultural sector. The shifts in temperature, rainfall patterns, and the frequency of extreme weather events have significant impacts on crop productivity [64]. VIs provide essential insights into plant response to varying climatic conditions, making them invaluable in climate-change studies as a powerful tool for studying climate-change effects on agriculture and forestry. Moreover, VIs are correlated with climate variables such as precipitation and temperature, establishing them as significant indicators of climate-change effects on terrestrial ecosystems [9]. VIs, such as the normalized difference vegetation index (NDVI), have been used to monitor vegetation dynamics and their relationship with climate factors such as precipitation [30]. For example, a significant correlation was found between the NDVI, temperature, and precipitation trends in the Tibetan Plateau, highlighting the potential of the NDVI as an indicator of climate change [31]. Moreover, the NDVI has been widely used to track climate-change impacts on crop phenology and productivity [32] and has been used as a proxy to assess changes in plant phenology and productivity in response to climate change [13]. The enhanced vegetation index (EVI), which is sensitive to canopy structural variations, has played a pivotal role in climate-change studies by being utilized to assess the effects of climate change in the Amazon Basin, thereby furnishing scientific evidence of climate change's impact and supplying information to local government bodies that is instrumental in shaping policy decisions geared towards the protection of the Amazon Basin [33]. Other indices like the SAVI has proven to be an effective index for mapping soil moisture in cultivated agricultural areas that range from bare soil to dense vegetation, aiding in mitigating the water crisis in arid and semi-arid regions worldwide, a situation expected to intensify due to the rapidly increasing human population and the evolving global climate [34]. In addition, air pollution, a key factor linked to climate change that has an impact on people's quality of life and can be controlled through vegetation [65], can distort the precision of several vegetation indices in areas with considerable atmospheric aerosol presence, underlining the critical importance of accounting for atmospheric particulate pollution in such evaluations [35].

### 4.2. Organic Production

Organic agriculture is posited as a viable alternative for mitigating the environmental repercussions of agronomic production while bolstering biodiversity. One of the salient

constraints in organic agriculture is weed proliferation, the management of which is crucial, yet challenging, for ensuring optimal yields. Through NDVI cluster analysis, the spatial and temporal heterogeneity of crops can be discerned in the early stages, thereby furnishing strategic insights for agricultural management geared towards enhancing the yield quality in organic production [36,37]. Furthermore, the incorporation of deleterious chemicals, particularly in the form of pesticides, in agricultural production poses substantial risks to consumers. Such chemicals are not permissible in organic farming practices. The NDVI offers the capacity to detect phytochemicals or their concomitant effects, such as herbicides in organic agriculture, thereby facilitating adherence to certification audits and ensuring the maintenance of safe pesticide thresholds in conventional agricultural practices [38–40]. In a parallel vein, an investigation [41] was carried out to study the potential of the NDVI in augmenting the traceability of organic produce through the integration of machine learning. This is especially pertinent in the context of organic agriculture, which necessitates the adoption of non-chemical weed abatement, crop rotation, and sustainable fertilization and water-management protocols.

### 4.3. Disaster Management

The applicability of VIs in disaster management, especially in agricultural landscapes, is gaining attention. For instance, VIs such as the normalized difference water index (NDWI) [66], which has been specifically developed for remote sensing of vegetation liquid water from space, have proven effective in detecting flood-affected areas [42], while the burn area index (BAI), has been used to identify areas affected by wildfires [43]. The quick and accurate detection of disaster-stricken regions allows for efficient disaster response and recovery, minimizing the impact on agricultural productivity. The crop water stress index (CWSI), initially developed for irrigation scheduling, has been used to predict drought conditions, proving to be a reliable indicator for assessing and monitoring regional droughts [44]. The ability to forecast droughts can guide mitigation measures, minimizing crop losses. Besides, the difference in vegetation response before and after a disaster, captured by indices like the NDVI and EVI, can be used for rapid damage assessment post-disaster [45]. This application extends to the forestry sector where VIs aid in monitoring and mapping wildfire damages and post-fire recovery [46].

### 4.4. Microorganisms and Yeasts

Interestingly, the use of VIs is not limited to large-scale agricultural and forest monitoring. Recent research has shown the potential of these indices in studying microorganisms and yeasts that have significant impacts on agricultural productivity. For instance, the NDVI has been used to detect fungal infections in crops, a challenging task due to the minute scale of the pathogens [47,48]. Moreover, it can function as a marker for measuring plant weight and the commencement of enzymatic activities linked to the induction of plant defense mechanisms [49]. Ref. [50] studied the interaction of climate, topography, and soil properties with cropland using MODIS-NDVI product data and machine learning methods. Ref. [51] employed fluorescence, thermography, and NDVI techniques in lettuce to identify the presence of *Rhizoctonia solani* infection, revealing that in certain instances, infected plants were detected prior to the manifestation of visible symptoms. Similarly, the use of VIs in detecting yeasts, particularly those involved in fermentation processes in the wine industry, has been explored. Yeasts, essential to the fermentation process, significantly influence the taste and quality of the wine. Ref. [52] have indicated that satellite multispectral imaging, through the NDVI, could potentially be used to monitor microbial terroir and yeast species richness within the vineyards. This result was confirmed in a subsequent study, finding that satellite imagery can differentiate yeasts according to their fermentative capacity and could serve as a valuable asset for managing wine differentiation and a decision-making resource for designation of origin (DO) regulators and viticulturists [53]. These novel applications could offer a non-invasive method for winemakers to improve knowledge of the microbial status of the local terroir, thus improving product distinction,

quality, and added value. Furthermore, accurate assessment of soil conditions and crop health is crucial for identifying and addressing issues like pollution and soil-borne diseases. VIs can be helpful in this task and images have the potential of detecting changes in soil microbial community composition. Ref. [54] found in winter wheat that hyperspectral reflectance can detect the presence of aggressive soil pathogens and their patterns differed significantly based on geographical distance and microbial species loss. Moreover, they found a positive correlation between the NDVI and bacterial species richness.

### 4.5. Quality Assessment

Although VIs have traditionally been used to monitor crop and forest vegetation, other traditional uses were the assessment of the quality of the production. Quality assessment through remote sensing is instrumental in monitoring plant stress and status, which impacts several aspects of fruit or grain production such as weight, composition, or nutrient levels. Thus, Ref. [55] used RGB drone imagery to correlate VIs with wine-quality variables, finding significant correlations and suggesting that conventional digital imagery can enhance wine production management and productivity by providing insights into critical quality variables. In addition, Ref. [16] found significant differences between vigor and quality parameters in grapes and also in wine. Finally, several relationships between VIs and crop quality have been identified not only in woody crops but also in other types of crops such as cereals [56].

### 4.6. Leaf Area and Photosynthetically Active Radiation (PAR)

Leaf area, photosynthetically active radiation (PAR), FPAR (fraction of PAR), and FA-PAR (fraction of absorbed PAR by a vegetation canopy) are critical biophysical parameters of crops and ecosystems and, although they are not VIs in the strictest sense, rely on similar principles and can be estimated using VIs [67]. FAPAR and FPAR are similar but slightly different. Both are measures used in vegetation studies, but FAPAR represents the fraction of incoming photosynthetically active radiation absorbed by vegetation, indicating its photosynthetic ability, and FPAR represents the fraction of incoming photosynthetically active radiation that is intercepted by vegetation. FPAR and the NDVI have a clear relationship that is independent of pixel heterogeneity [68]. On the other hand, Ref. [57] showed a strong linear relationship between the satellite-derived NDVI time series and the leaf area of the crop, suggesting that satellite NDVI can effectively detect the evolution of the NDVI within a crop. Therefore, VIs are instrumental in the calculation of leaf area index (LAI), a common index to evaluate the leaf area of the plant and closely related to FAPAR [69,70]. The LAI, which represents the total one-sided area of leaf tissue per ground surface area, provides information about canopy structure and plant vigor. Methods that determine the radiation intercepted by the plant to estimate the LAI have been developed [59]. Ref. [71] found that the correlation between the NDVI and LAI can change throughout different seasons and from year to year, aligning with the fluctuations in the phenological growth of trees and in response to temporal changes in environmental factors. Therefore, it is possible to establish a relationship between the leaf area and the VIs, such as the NDVI. In this sense, Ref. [58] evaluated various methods for estimating corn LAI in northeastern China using hyperspectral reflectance data, computing several indices like the NDVI and EVI. Ref. [60] employed satellite-derived NDVI and proposed a regional-scale method for accurately estimating rice LAI during the growing period, and Ref. [61] focused their research in the semi-arid grassland of Inner Mongolia, formulating specific equations to estimate the leaf area index (LAI) within the typical vegetation range observed during the growing season. However, these equations vary from one area to another, and it is very important to select the appropriate NDVI-LAI equation, or a combination of equations, based on the trade-off between the scale of the research and the availability of data [62]. Nevertheless, the NDVI suffers from saturation at high density canopies and recently other VIs have been proposed to improve LAI estimations [63].

## 5. Limitations, Challenges, and Future Directions

While VIs offer a multitude of applications in agriculture and forestry, their use is not without limitations and challenges. As discussed, these indices have been utilized in a range of areas, from climate-change studies, organic production, and disaster management to the assessment of microorganisms and yeasts. However, the extraction of accurate and useful information from these indices requires addressing several obstacles.

### 5.1. Technical Hurdles

One of the major challenges relates to technical issues in data acquisition and processing. Data quality can be affected by factors such as atmospheric conditions, sensor calibration, and sun angle, which may lead to inaccuracies in vegetation index values [72] and misalignments between several sources of data [73]. Furthermore, most VIs are sensitive to soil background effects, particularly in areas with sparse vegetation [74]. Addressing these technical issues requires advanced image-processing techniques and robust algorithms for atmospheric correction and soil background adjustment.

### 5.2. Interpretational Challenges

Interpretation of vegetation index values can also be challenging. Variations in VIs do not always reflect changes in vegetation health or coverage but may be influenced by factors such as soil moisture, atmospheric conditions, or the presence of non-vegetated objects within the pixel [75]. Furthermore, when using a single image, the challenge lies solely in interpreting spatial patterns of the vegetation index or selecting the appropriate VI [76]. However, when multiple images of the same area are used over time, the difficulty exponentially increases as specific methods need to be employed to interpret temporal patterns [77]. In addition, new applications such as the use of remote sensing to estimate the ecological quality of the environment or microbial terroir will require new forms of interpretation.

### 5.3. Data-Quality Issues

Data quality is another significant concern. Satellite-derived data, commonly used for calculating VIs, can be affected by cloud cover, sensor noise, or temporal-resolution limitations, leading to gaps or inaccuracies in the data [78]. These issues need to be carefully considered when using VIs for decision-making or scientific research. Despite these challenges, there are several ways these issues can be overcome. For example, the use of multi-sensor data can help mitigate data-quality issues related to cloud cover or sensor noise [79].

Looking ahead, the field of precision agriculture for woody crops is poised for significant advancements. The development of more sophisticated VIs that can account for the unique attributes of woody crops, for example, coupled with the integration of other data types such as soil moisture and weather data, will further enhance our ability to monitor and manage these crucial crops. Since the 1970s, VIs have evolved with advancements in satellite sensor technology and spectroscopy, driven by new scientific demands. Today, the integration of VIs with other precision agriculture technologies, such as drone-based imaging and on-the-ground sensors, will provide a more comprehensive picture of crop health and needs. This holistic approach to crop management will drive the sustainability and profitability of woody-crop farming in the years to come. The selection of VIs should consider the spectral sensitivity and suitability of the application, because as indicated, the ability of VIs to obtain valuable information is influenced by various factors such as the vegetation itself or pollution. Analytical steps should verify VI results using in situ data (ground-truth data), and base conclusions on multiple indicators. Future VIs, expected to have improved signal-to-noise ratios, will leverage new satellite missions and emerging vegetation metrics [75], therefore, creating new opportunities to explore further uses of these indices, as exemplified by the applications studied in this review.

The future trends in the use of VIs are likely to be influenced by advances in remote-sensing technology and data analytics. As sensor technology improves, new indices may be developed that are more resistant to noise or more sensitive to changes and background effects [80]. Additionally, the integration of VIs with other data types, such as LiDAR, thermal, or radar data, may provide more comprehensive information about vegetation and environmental conditions [81,82].

This review brings to the forefront topics that are typically not explored by mainstream studies using VIs (such as climate change, quality assessment, disaster management, or the study of microorganisms and yeasts). From this review, it is clear that these indices have applications that go beyond the simple monitoring of vegetation, which is what they were originally designed for. By shedding light on these alternative uses, this review aims to expand the scope and understanding of the potential and versatility of VIs in various fields. Therefore, further research is needed in areas other than vegetation monitoring to expand the potential of VIs. Additionally, research should focus on developing and validating new indices that are more robust to the limitations of current indices. Likewise, further research in the field of yeast and microorganism assessment could revolutionize soil-health assessment practices, potentially making them faster, more efficient, and less invasive.

## 6. Conclusions

Vegetation indices (VIs) have been used in many studies related to agriculture and farming systems, forestry, climate change, disaster management, microorganisms, and yeasts. These applications demonstrate the versatility of VIs and their potential for use in a wide range of fields. In agriculture and forestry, VIs provide a non-invasive and cost-effective way to monitor plant growth and health, as well as to assess the impact of environmental factors on vegetation. The application of these indices, ranging from the most commonly used vegetation index, the normalized difference vegetation index (NDVI), to the more specialized canopy chlorophyll content index (CCCI), has seen a proliferation into numerous related and tangential research areas.

The versatility and potential of VIs remain evident. The broad range of applications emphasizes the vast potential of these indices to contribute to our understanding and management of agricultural and forestry systems. Moving forward, continued exploration and adaptation of VIs will undoubtedly yield even more profound insights and practical applications, solidifying their role in securing a sustainable future for agriculture and forestry. Nevertheless, the use of VIs extends well beyond their traditional roles in assessing vegetation health. Their potential applications in areas such as climate-change research, disaster management, and microbial detection represent the untapped opportunities for utilizing these indices to better understand and manage our environment. While the exploration of these alternative applications is still in its early stages, the initial findings are promising, pointing to a future where VIs will be fundamental tools in a variety of scientific and practical disciplines.

Still, the evolution and expanding use of VIs, however, is not without its challenges. As identified in the review, technical hurdles, interpretational challenges, and issues with data quality present significant obstacles to the wider adoption and efficacy of these indices. In such a way, VIs pose challenges when used for conventional applications like vegetative assessment. However, these challenges become even more pronounced when attempting to apply new ideas, such as assessing microbial terroir. Nonetheless, these obstacles are not insurmountable. With advancements in technology and ongoing research, VIs will continue to be a vital tool not only in agriculture and forestry, but also in novel, innovative, and out-of-the-box applications, providing valuable insights for decision-making and scientific research.

**Author Contributions:** Conceptualization, R.M.-P., S.V. and D.C.; methodology, R.M.-P., S.V. and D.C.; software, R.M.-P., S.V. and D.C.; validation, S.V. and D.C.; formal analysis, R.M.-P., S.V. and D.C.; investigation, S.V.; resources, R.M.-P., S.V. and D.C.; data curation, S.V.; writing—original draft preparation, S.V.; writing—review and editing, R.M.-P. and D.C.; visualization, R.M.-P.; supervision, D.C. All authors have read and agreed to the published version of the manuscript.

**Funding:** This research received no funding.

**Conflicts of Interest:** The authors declare no conflict of interest.

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
