# Peer review of "Beyond Vegetation: A Review Unveiling Additional Insights into Agriculture and Forestry through the Application of Vegetation Indices"

_2571-8800, doi:10.3390/j6030028_

Round 1
Reviewer 1 Report
This is a well-written piece of work, with good flow and very good use of English language. There is a couple of typos, but little to change.
It is not a comprehensive review (remove “comprehensive” from lines 108 and 128; it would take many more pages and references to make it comprehensive), but I do not think this was the aim. It reads a lot like a literature review of a potential research topic (PhD or postdoc) on vegetation indices.
Still, in my opinion, it is good enough for publication with minor changes.
Lines 26 and 93. Please remove text “thinking outside the box”. It is unnecessary.
Lines 111-112: It seems the “…four key research areas…” are missing from the text (after semicolon). Please add, or correct accordingly.
Line 162. “stablishing”? Perhaps “establishing”?
Line 273. Remove “Thus”.
Line 382. What is FAPAR? Please expand the acronym. [FPAR is expanded, but not FAPAR]
Line 398. “stablish”? Do you mean “establish”?
General comment: In some parts of the text, the first person is used (e.g. “we” in lines 86-87). I personally prefer third person or passive voice. I would leave it on the editor to decide on this.
Author Response
Dear Reviewer,
We would like to thank you for reviewing our manuscript. According to the feedback provided, the manuscript has been revised and improved to enhance readability and better understanding. We have included a point-by-point response (in red) to each comment. We hope you consider positively the effort we made. Please do not hesitate to contact us if there are any other suggestions.
Thank you for your consideration.
Reviewer 1
This is a well-written piece of work, with good flow and very good use of English language. There is a couple of typos, but little to change.
We thank the reviewer for their kind words.
It is not a comprehensive review (remove “comprehensive” from lines 108 and 128; it would take many more pages and references to make it comprehensive), but I do not think this was the aim. It reads a lot like a literature review of a potential research topic (PhD or postdoc) on vegetation indices.
True, our intention was to show the potential areas of application of vegetation indices beyond vegetation. The term “comprehensive” has been deleted.
Still, in my opinion, it is good enough for publication with minor changes.
Lines 26 and 93. Please remove text “thinking outside the box”. It is unnecessary.
These words have been deleted.
Lines 111-112: It seems the “…four key research areas…” are missing from the text (after semicolon). Please add, or correct accordingly.
Corrected.
Line 162. “stablishing”? Perhaps “establishing”?
Corrected
Line 273. Remove “Thus”.
Corrected
Line 382. What is FAPAR? Please expand the acronym. [FPAR is expanded, but not FAPAR]
Corrected
Line 398. “stablish”? Do you mean “establish”?
Corrected
General comment: In some parts of the text, the first person is used (e.g. “we” in lines 86-87). I personally prefer third person or passive voice. I would leave it on the editor to decide on this.
We have changed all subjects to third person or passive voice.

Reviewer 2 Report
The review concerned an article entitled: Beyond vegetation: Unveiling additional insights in agriculture 2 and forestry through the application of vegetation indices. Vegetation indices (VIs) have played a crucial role in monitoring plant growth, assessing vegetation health, and understanding the impact of environmental factors on vegetation. The article explores both the traditional and novel applications of VIs in various fields. While the primary focus has been on agriculture and forestry, VIs have found utility in other areas such as climate change studies, disaster management, and microorganism assessment. Despite their potential, the practical application of VIs presents certain challenges that need to be addressed. The objective of this research is not only to assess the current state of VIs but also to explore innovative and unconventional applications, thinking outside the box. Rather than solely following mainstream approaches, it is important to anticipate the potential of this dynamic and rapidly advancing scientific tool. By embracing the possibilities offered by VIs, we can unlock new insights and make significant strides in various fields.
In my opinion the article is worth consideration for publication.
Some answers for the questions – which should in my opinion included in the introduction:
What are some tools used in remote sensing?
Besides reading vegetation characteristics, what other parameters can be extracted using remote sensing instruments?
What is the aim of the burgeoning field of study mentioned in the text?
Which areas are mentioned as contexts where VIs can provide valuable information?
What is the objective of this review regarding the exploration of VIs?
How does the article encourage researchers, practitioners, and policymakers regarding VIs?
The methodology described in the text is comprehensive and systematic, focusing on different aspects of vegetation indices and their applications. However, there are a few potential improvements or additional considerations that could enhance the methodology:
Selection criteria and inclusion/exclusion criteria: It would be beneficial to outline specific criteria for selecting studies from the search results. This could involve defining inclusion and exclusion criteria based on relevance, study quality, publication date, or other factors to ensure a more focused and representative analysis.
SSearch strategy transparency: Providing a detailed description of the search strategy, including the specific keywords, search operators, and combinations used for each search query, would enhance the transparency and replicability of the study.
Data extraction and analysis: Describing the process for extracting relevant data from the selected studies and outlining the methods for data analysis would provide clarity on how the information was synthesized and interpreted.
Line 296. The phrase "[35] investigated the effectiveness of several VIs in assessing vegetation in regions with high atmospheric aerosol presence, highlighting the importance of taking into account the atmospheric particulate pollution." It is unclear what specific findings or insights were highlighted by the investigation mentioned. Additional details or explanation would provide better understanding. Authors should read and add the info from plant vs pollution studies – eg. https://doi.org/10.3390/su15097568 or https://doi.org/10.1016/j.ecolind.2023.110259
Line 449. The statement "The selection of VIs should consider spectral sensitivity and application suitability." It would be helpful to provide more information on what factors contribute to the spectral sensitivity and application suitability of VIs. What specific considerations should be taken into account when selecting VIs for different purposes?
The sentence "Analytical steps should mitigate artefacts, verify VI results using in situ data, and base conclusions on multiple indicators." It is not clear what specific artefacts are being referred to and how they should be mitigated. Additionally, further elaboration on the process of verifying VI results using in situ data would provide a better understanding of the methodology.
The mention of "new opportunities to further investigate new applications of indexes, such as those indicated in this article." It is unclear which specific indexes are being referred to and what potential new applications are being suggested. Providing more details or examples would enhance clarity.
The statement "This review brings to the forefront topics that are typically not explored by mainstream studies using VIs." It would be beneficial to provide specific examples of these topics that are not typically explored and how they expand the understanding and potential of VIs in various fields. In example the monitoring of microplactic – or remediation of microplastic by trees eg. doi.org/10.3390/plants12030462
The phrase "More studies are required to understand the relationship between VIs and various biophysical parameters under different environmental conditions." It is not clear which specific biophysical parameters are being referred to and how VIs can be used to understand their relationship. Further clarification on this point would be helpful.
Author Response
Dear Reviewer,
We would like to thank you for reviewing our manuscript. According to the feedback provided, the manuscript has been revised and improved to enhance readability and better understanding. We have included a point-by-point response (in red) to each comment. We hope you consider positively the effort we made. Please do not hesitate to contact us if there are any other suggestions.
Thank you for your consideration.
Reviewer 2
The review concerned an article entitled: Beyond vegetation: Unveiling additional insights in agriculture 2 and forestry through the application of vegetation indices. Vegetation indices (VIs) have played a crucial role in monitoring plant growth, assessing vegetation health, and understanding the impact of environmental factors on vegetation. The article explores both the traditional and novel applications of VIs in various fields. While the primary focus has been on agriculture and forestry, VIs have found utility in other areas such as climate change studies, disaster management, and microorganism assessment. Despite their potential, the practical application of VIs presents certain challenges that need to be addressed. The objective of this research is not only to assess the current state of VIs but also to explore innovative and unconventional applications, thinking outside the box. Rather than solely following mainstream approaches, it is important to anticipate the potential of this dynamic and rapidly advancing scientific tool. By embracing the possibilities offered by VIs, we can unlock new insights and make significant strides in various fields.
In my opinion the article is worth consideration for publication.
We thank the reviewer for their kind words.
Some answers for the questions – which should in my opinion included in the introduction: What are some tools used in remote sensing? Besides reading vegetation characteristics, what other parameters can be extracted using remote sensing instruments? What is the aim of the burgeoning field of study mentioned in the text? Which areas are mentioned as contexts where VIs can provide valuable information? What is the objective of this review regarding the exploration of VIs? How does the article encourage researchers, practitioners, and policymakers regarding VIs?
The introduction has been rewritten to address the comments of the reviewer (information added in lines 41-44, 47-51, 65-67, 80-82, 92-94, 105-108).
The methodology described in the text is comprehensive and systematic, focusing on different aspects of vegetation indices and their applications. However, there are a few potential improvements or additional considerations that could enhance the methodology:
Selection criteria and inclusion/exclusion criteria: It would be beneficial to outline specific criteria for selecting studies from the search results. This could involve defining inclusion and exclusion criteria based on relevance, study quality, publication date, or other factors to ensure a more focused and representative analysis.
Information added (line 149-152):
“In order to maintain a reasonable volume of information, for each category the first 100 papers were analyzed, first by title and, if relevant for the interests of this review, the abstract and finally the full text, including references to try to find unconventional applications for VI”
SSearch strategy transparency: Providing a detailed description of the search strategy, including the specific keywords, search operators, and combinations used for each search query, would enhance the transparency and replicability of the study.
This information is included in section “2. Methodology”:
“The study is a review of literature sourced from the Scopus database, one of the most extensive bibliographic databases for research literature. The research methodology is designed to ensure a systematic and in-depth analysis of the selected studies. The process was initiated by focusing on four key research words associated with re-mote sensing and vegetation indices (VIs). For this, four distinct search queries were deployed: i) "Remote" AND "Sensing", ii) "Vegetation" AND "Index", iii) "satellite" AND "vegetation" AND "index", and iv) "drone" OR "uav" AND "vegetation" AND "index". The results from this first step of the search were presented in Figure 1. Next, attention was turned to the Normalized Difference Vegetation Index (NDVI) and searched the Scopus database exclusively for entries related to “NDVI”. The results of this search are presented separately in Figure 2. Further refining the investigation, a delve was made into other common VIs, leading to a search on i) “EVI”, ii) “SAVI”, iii) “NDRE”, and iv) “CWSI”. The findings from these searches are illustrated in Figure 3. Lastly, to understand the wider implications of NDVI beyond vegetation, specific searches were carried out linking NDVI with other relevant domains. These search strings included: i) "NDVI" AND "vegetation", ii) "NDVI" AND "climate" AND "change", iii) "NDVI" AND "disaster", and iv) "NDVI" AND "microbial". The results of these tar-geted searches are documented in Figure 4.”
Data extraction and analysis: Describing the process for extracting relevant data from the selected studies and outlining the methods for data analysis would provide clarity on how the information was synthesized and interpreted.
Information added (line 149-152):
“In order to maintain a reasonable volume of information, for each category the first 100 papers were analyzed, first by title and, if relevant for the interests of this review, the abstract and finally the full text, including references to try to find unconventional applications for VI”
Line 296. The phrase "[35] investigated the effectiveness of several VIs in assessing vegetation in regions with high atmospheric aerosol presence, highlighting the importance of taking into account the atmospheric particulate pollution." It is unclear what specific findings or insights were highlighted by the investigation mentioned. Additional details or explanation would provide better understanding. Authors should read and add the info from plant vs pollution studies – eg. https://doi.org/10.3390/su15097568 or https://doi.org/10.1016/j.ecolind.2023.110259
One of the references has been added and the paragraph was rewritten:
“In addition, air pollution, a key factor linked to climate change that has an impact on people's quality of life and can be controlled through vegetation [65], can distort the precision of several vegetation indices in areas with considerable atmospheric aerosol presence, underlining the critical importance of accounting for atmospheric particulate pollution in such evaluations [35].”
Line 449. The statement "The selection of VIs should consider spectral sensitivity and application suitability." It would be helpful to provide more information on what factors contribute to the spectral sensitivity and application suitability of VIs. What specific considerations should be taken into account when selecting VIs for different purposes?
It refers to the limitations indicated throughout the revision. The paragraph has been rewritten and some examples have been added:
“The selection of VIs should consider the spectral sensitivity and suitability of the application, because as indicated, the ability of VIs to obtain valuable information is in-fluenced by various factors such as the vegetation itself or pollution.”
The sentence "Analytical steps should mitigate artefacts, verify VI results using in situ data, and base conclusions on multiple indicators." It is not clear what specific artefacts are being referred to and how they should be mitigated. Additionally, further elaboration on the process of verifying VI results using in situ data would provide a better understanding of the methodology.
“In situ data” refers to “ground-truth”. The references to artifacts have been deleted to avoid confusion. The sentence has been rewritten:
“Analytical steps should verify VI results using in situ data (ground-truth data), and base conclusions on multiple indicators”
The mention of "new opportunities to further investigate new applications of indexes, such as those indicated in this article." It is unclear which specific indexes are being referred to and what potential new applications are being suggested. Providing more details or examples would enhance clarity.
It refers to the indices studied in this article (EVI, SAVI, NDRE, CWSI). The sentence has been rewritten:
“therefore, creating new opportunities to explore further uses of these indices, as exemplified by the applications studied in this review.”
The statement "This review brings to the forefront topics that are typically not explored by mainstream studies using VIs." It would be beneficial to provide specific examples of these topics that are not typically explored and how they expand the understanding and potential of VIs in various fields. In example the monitoring of microplactic – or remediation of microplastic by trees eg. doi.org/10.3390/plants12030462
Refers to all the topics already discussed in the review (Climate change, Organic production, Disaster Management, Microorganisms and Yeasts, Quality assessment, Leaf Area and Photosynthetically Active Radiation (FPAR) Calculation). The sentence has been rewritten to reference to the topics already mentioned:
“This review brings to the forefront topics that are typically not explored by mainstream studies using VIs (such as climate change, quality assessment, disaster management or the study of microorganisms and yeasts)”
The phrase "More studies are required to understand the relationship between VIs and various biophysical parameters under different environmental conditions." It is not clear which specific biophysical parameters are being referred to and how VIs can be used to understand their relationship. Further clarification on this point would be helpful.
The sentences has been deleted because it was redundant with the previous one (“Therefore, further research is needed in other areas different than vegetation monitoring to expand the potential of VIs”)
